# 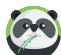 DADA: Dialect Adaptation via Dynamic Aggregation of Linguistic Rules

**Yanchen Liu** 🛡️  **William Held** 🐝  **Diyi Yang** 🌲

🛡️Harvard University, 🐝Georgia Institute of Technology, 🌲Stanford University
yanchenliu@g.harvard.edu, wheld3@gatech.edu, diyiy@cs.stanford.edu

## Abstract

Existing large language models (LLMs) that mainly focus on Standard American English (SAE) often lead to significantly worse performance when being applied to other English dialects. While existing mitigations tackle discrepancies for individual target dialects, they assume access to high-accuracy dialect identification systems. The boundaries between dialects are inherently flexible, making it difficult to categorize language into discrete predefined categories. In this work, we propose DADA (Dialect Adaptation via Dynamic Aggregation), a modular approach to imbue SAE-trained models with multi-dialectal robustness by composing adapters which handle specific linguistic features. The compositional architecture of DADA allows for both targeted adaptation to specific dialect variants and simultaneous adaptation to various dialects. We show that DADA is effective for both single task and instruction finetuned language models, offering an extensible and interpretable framework for adapting existing LLMs to different English dialects.[1]

## 1 Introduction

As Natural Language Processing (NLP) becomes even more impactful, the equitable distribution of its benefits becomes an increasing concern. Specifically, NLP tooling is often trained and evaluated on dominant language variants, such as Standard American English (SAE). This results in a significant decline in the performance when these tools are applied to non-SAE dialects. Studies have revealed that SAE models tested on African American Vernacular English (AAVE) encounter difficulties in language identification (Jurgens et al., 2017a) as well as various other natural language tasks (Jørgensen et al., 2016a; Kiritchenko and Mohammad,

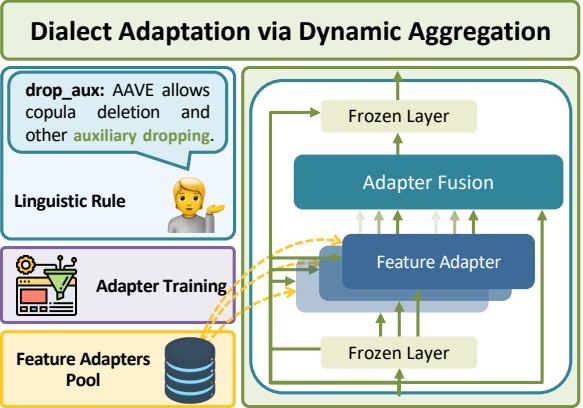

Figure 1: DADA dynamically composes adapters that handle specific features of dialectal variation to adapt an SAE model to various dialects by leveraging their commonality. We train nearly 200 feature adapters to capture the linguistic differences between SAE and its dialect variants. These feature adapters can be composed flexibly and arbitrarily to target different dialects.

2018; Blodgett et al., 2018). These challenges extend to automated speech recognition used by virtual assistants (Koenecke et al., 2020) and hate speech detection employed by online media platforms (Davidson et al., 2019; Sap et al., 2019; Rios, 2020; Mozafari et al., 2020; Halevy et al., 2021a; Zhou et al., 2021). Notably, even large language models are not exempt from these limitations (Bommasani et al., 2021; Solaiman and Dennison, 2021; Rae et al., 2022; Liang et al., 2022). Such performance disparities raise ethical and moral concerns regarding the potential for racial disparities in the seemingly expeditious development of language technologies (Hovy and Spruit, 2016; Blodgett and O'Connor, 2017; Halevy et al., 2021b).

Existing research to mitigate this disparity has mainly focused on dialectal adaptation targeting individual dialects of interest (Ziems et al., 2022; Garcia and Firat, 2022; Ziems et al., 2023; Sun et al., 2022). This approach is a powerful first step, but it has key limitations of missing connectedness

---

[1]All the code, synthetic datasets, and trained adapters in this work are available at https://github.com/SALT-NLP/DADA.

among dialects; for instance, English alone has 77 recognized variants that vary internally (Koenecke et al., 2020; Demszky et al., 2021). Prior adaptation methods also require highly accurate dialect identification systems for real-world uses, leading to the development of separate systems for different dialects. Such separate systems are not yet available for many dialects and related languages (Malmasi et al., 2016; Aepli et al., 2023; Chakravarthi et al., 2021; Aepli et al., 2022). Alternative approaches train models using a combination of various dialect variants in a multi-task learning manner (Caruana, 1997; Liu et al., 2019a). However, this approach requires training new models for dialectal NLP from scratch simultaneously with data from all desired dialects. This training process is prohibitive, especially given the trend towards larger language models with costs upwards of millions of dollars[2]. Thus, there is a pressing need for an effective and extensible approach that can adapt existing models to the multi-dialectal setting.

Previous linguistic works have developed a collection of lexical and morphosyntactic features that describe the differences between SAE and various other English dialects (Kortmann et al., 2020; Ziems et al., 2023). Many dialects can be described by this common set of features or linguistic rules, with each dialect expressing a subset of the feature space. In addition, dialects are not deterministic speech patterns but rather ranges of acceptable use of these features that speakers adjust based on social contexts (Ziems et al., 2023; Koenecke et al., 2020; Demszky et al., 2021). As a result, dialects do not neatly fit into predefined categories.

To this end, we develop a model which handles this reality by accommodating the diversity of English variants at a fine-grained level (linguistic features or linguistic rules). Concretely, we propose Dialect Adaptation via Dynamic Aggregation (DADA): a modular approach to adapt an established model trained on SAE to dialect variants by composing linguistic features. DADA captures and encapsulates each feature using adapters (Houlsby et al., 2019) trained on individual feature rules. Feature adapters dynamically aggregate at test time using adapter fusion (Pfeiffer et al., 2021), which enables the SAE model to flexibly adapt to dialects. The modular design of DADA enables targeted adaptation to specific dialect variants or simulta-

neous adaptation to multiple dialects. As a result of its compositional nature, DADA also makes it easy to re-use feature adapters regardless of dialect, speaker, or time variations in feature usage. The modular architecture ensures interpretability by enabling analysis of the components responsible for the improvement in performance.

To sum up, our work contributes the following:

- We propose a modular approach DADA to adapt the standard SAE model to dialect variants via a dynamic aggregation of different linguistic features. (Sec. 3)

- We train nearly 200 feature adapters, which can be flexibly composed to target different dialects. Moreover, we demonstrate that DADA with all the trained feature adapters can consistently improve model performance across five English dialects. (Sec. 4)

- DADA exhibits strong interpretability. Using AAVE as an example, we illustrate that DADA possesses the capability to detect the relevant linguistic features for a given input and subsequently activate the corresponding feature adapters. (Sec. 5)

- We show that DADA improves dialectal robustness in task-agnostic instruction-tuned LLMs using FLAN-T5 (Chung et al., 2022) (Sec. 6), which highlights the capability of DADA in learning task-agnostic features that can be applied to newer general-purpose models.

## 2 Related Work

**Dialect**  NLP research tends to focus primarily on dominant dialects represented in "textbook" grammar, such as Standard American English (SAE), over lower-resource dialects. The performance disparity in resulting models is pervasive (Koenecke et al., 2020; Davidson et al., 2019; Sap et al., 2019; Rios, 2020; Mozafari et al., 2020; Halevy et al., 2021a; Zhou et al., 2021; Ziems et al., 2022, 2023; Sun et al., 2022). The existence of such performance disparities raises ethical and moral concerns where NLP can potentially exacerbate the marginalization of the speakers of these dialects (Blodgett and O'Connor, 2017; Halevy et al., 2021b). Lacking a common dialectal evaluation, NLP can reinforce existing power discrepancies (Hovy and Spruit, 2016; Bommasani et al., 2021). Existing works on English dialects have mainly focused on adapting models to individual dialects, such as

---

[2] https://lambdalabs.com/blog/demystifying-gpt-3

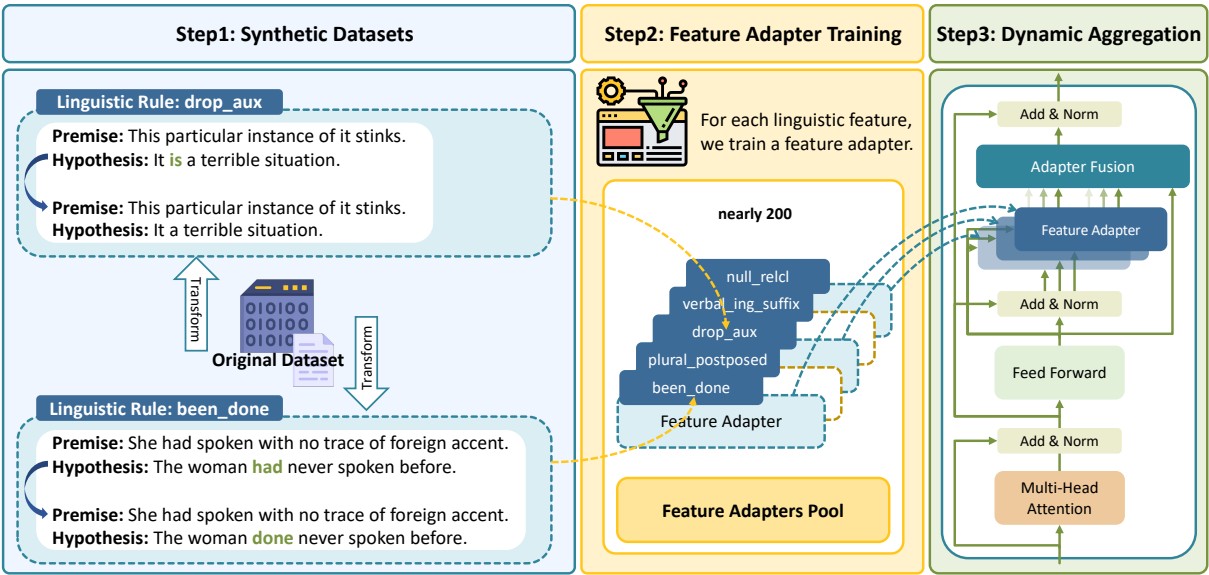

Figure 2: The overall process of DADA. We first construct a synthetic dataset $D_i$ by applying each linguistic transformation rule $T_i \in \mathcal{T}$, such as **drop_aux**: *"AAVE allows copula deletion and other auxiliary dropping"*, to each individual training example within the original training dataset $D$ (taking MNLI as an example). Then we develop a feature adapter $A_i$ for each linguistic rule $T_i$ by training it on the corresponding synthetic dataset $D_i$. We select the backbone model trained on the original **SAE** task datasets to enable the feature adapter to capture linguistic differences while disregarding the task-specific information.

African American Vernacular English (AAVE) (Jør-gensen et al., 2016b; Blevins et al., 2016; Ziems et al., 2022). However, for real-world use, such systems would require another system to recognize these dialects so that the appropriate model can be used for each input. This task itself is challenging, with state-of-the-art systems showing relatively low accuracy even when distinguishing high-resource dialects of English (Zampieri et al., 2023). Our work avoids this flaw by modeling multiple dialects at once using multidialectal training data. Multidi-alectal training data has been shown to potentially increase robustness across all dialects in multiple prior works around data collection (Jurgens et al., 2017b) and augmentation (Ziems et al., 2023).

**Parameter-Efficient Learning** To efficiently transfer pretrained language models to downstream tasks, several techniques (He et al., 2022) have been proposed to update only a small number of extra parameters while keeping most parameters frozen. For example, adapter tuning (Houlsby et al., 2019; Pfeiffer et al., 2020) adapts models using small bottleneck modules. Prefix tuning (Li and Liang, 2021) and prompt tuning (Lester et al., 2021) prepend additional tunable prefix tokens to input or hidden layers. Brown et al. (2020); Liu et al. (2022a,b) prompt language models for specific tasks without any parameter updates by in-context

learning. Besides, several research efforts have been carried out to ensemble parameter-efficient components for multi-task learning and domain adaptation. Pfeiffer et al. (2021) propose to aggregate adapters trained on source tasks with an attentional layer to transfer acquired knowledge to a target task. Asai et al. (2022) introduce a sim-ilar method, while using soft prompts instead of adapters. To improve robustness against dataset shortcuts, Liu et al. (2023) combine adapters with gating networks. Recently, Held et al. (2023) use adapter stacking for task-agnostic robustness tar-geting individual dialects. However, given the in-herent flexibility of dialects, there arises a necessity for a method to enhance multi-dialectal robustness. Moreover, the existence of well-defined transfor-mation rules between dialects, which is uncommon in other domains, allows us to achieve finer-grained adaptation by aggregating linguistic features.

**Instruction Tuning** Inspired by the success of prompting LLMs to adapt to various tasks (Brown et al., 2020), instruction tuning (Sanh et al., 2022; Wei et al., 2022; Ouyang et al., 2022) propose to finetune language models on a variety of tasks de-scribed through instructions to achieve the multi-task capability and to enhance zero-shot perfor-mance on unseen tasks. Since instruction tuning in-volves prompting the language models at the input

level, our approach is orthogonal to it and can be employed in conjunction to enhance model's multi-task and multi-dialect abilities simultaneously.

## 3 DADA

We introduce Dialect Adaptation via Dynamic Aggregation (DADA), a modular method for adapting an existing model trained on the Standard American English (**SAE**) to accommodate dialect variants at a finer-grained level. Our proposed method deploys a dynamic aggregation of feature adapters, which characterize the divergence of linguistic features between **SAE** and its dialect variants. Specifically, DADA involves the creation of a synthetic training dataset for each individual feature using transformation rules (Ziems et al., 2023). These synthetic datasets are used to train respective adapters for each linguistic feature. Finally, we compose these feature adapters to create a single model via an additional fusion layer.

### 3.1 Synthetic Datasets

Previous works have discerned a series of linguistic divergences and devised `Multi-VALUE`, a collection of lexical and morphosyntactic transformation rules [3] between **SAE** and its 50 dialect variants (Ziems et al., 2022, 2023), including Appalachian English (**AppE**), Chicano English (**ChcE**), Colloquial Singapore English (**CollSgE**), Indian English(**IndE**), and African American Vernacular English (**AAVE**), among others. For instance, a well-known linguistic feature of **AAVE** is the use of `Negative Concord`, where two negative morphemes are employed to convey a single negation (Martin and Wolfram, 2021). This transformation rule is sensitive to the verb-object dependency structure and necessitates an indefinite noun object (Green, 2002). As an example, the **SAE** sentence *"He doesn't have a camera"* could be rendered as *"He don't have no camera"* in **AAVE**.

Let $\mathcal{T} = \{T_1, T_2, ...T_N\}$ denote the set of transformation rules between **SAE** and its dialect variants. For each transformation rule $T_i \in \mathcal{T}$, we can generate a corresponding synthetic dataset $D_i$ by applying the respective rule to each individual training example within the original training dataset $D$.

### 3.2 Feature Adapter

Adapter tuning is known for its ability to adapt quickly to new tasks without catastrophic forgetting (Pfeiffer et al., 2021). Given these benefits and the inherent modularity of adapters, we develop a feature adapter $A_i$ for each of the $N$ linguistic transformation rules $T_i \in \mathcal{T}$ by training it on the corresponding synthetic dataset $D_i$ created in Sec. 3.1. We insert an adapter module after each feed-forward layer [4] of the backbone model $M$ that has been trained on the original **SAE** task datasets, in order to target specific lexical and morphosyntactic differences between **SAE** and its dialect variants.

### 3.3 Dynamic Aggregation

In Sec. 3.2, we described the process of training feature adapter $A_i$ for each linguistic transformation rule to capture a specific type of linguistic difference between **SAE** and its dialect variants. However, it is common for multiple linguistic differences to co-occur within a single sentence in real-world scenarios, thereby necessitating the model to simultaneously consider these distinct linguistic features to varying degrees.

Therefore, we propose to dynamically aggregate the $N$ trained feature adapters, denoted as $\mathcal{A} = \{A_1, A_2, ...A_N\}$, into the **SAE**-trained backbone model $M$ via an additional fusion layer (Pfeiffer et al., 2021). For this purpose, we first construct a super-synthetic training dataset $\mathfrak{D}$, employing the same approach as described in Sec. 3.1, but with all lexical and morphosyntactic transformation rules $\mathcal{T} = \{T_1, T_2, ...T_N\}$ applied. After incorporating the $N$ trained feature adapters $\mathcal{A}$ and a fusion layer into each layer of the backbone model, we train the fusion layers using the super-synthetic training dataset $\mathfrak{D}$, while keeping the feature adapters $\mathcal{A}$ and the backbone model $M$ frozen.

Following Pfeiffer et al. (2021), we define the fusion layer as a composition of *Key*, *Value* and *Query* matrices at each layer $l$ of the transformer, denoted by $\mathbf{K}_l$, $\mathbf{V}_l$ and $\mathbf{Q}_l$ respectively. The output of the feedforward layer $\mathbf{h}_l$ is taken as the query vector and the output of each feature adapter $A_i$, denoted as $\mathbf{a}_{l,i}$ is used as input to both the value and key transformations. With this attention-like fusion layer (Vaswani et al., 2017), the outputs of

---

[3]See Appendix A for a detailed description of each transformation rule and the statistics of the corresponding synthetic training dataset.

[4]There are different implementation variants for adapter tuning, and in our work, we follow Pfeiffer et al. (2020) by only inserting adapter modules after each feed-forward layer, while in some other works, adapters are inserted after multi-head attention layers as well.

all feature adapters are combined as followed:

$$\mathbf{s}_l = softmax(\mathbf{h}_l^T \mathbf{Q}_l \cdot \mathbf{a}_{l,i}^T \mathbf{K}_l), i \in \{1, ..., N\},$$
$$\mathbf{a}_{l,i}' = \mathbf{a}_{l,i}^T \mathbf{V}_l, i \in \{1, ..., N\},$$
$$\mathbf{A}_l' = [\mathbf{a}_{l,0}', ... \mathbf{a}_{l,N}'],$$
$$\mathbf{o}_l = \mathbf{s}_l^T \mathbf{A}_l',$$

where $[\cdot, \cdot]$ indicates the concatenation of vectors and $\mathbf{o}_l$ is the output of the $l$-th fusion layer. Through training on the super-synthetic dataset $\mathfrak{D}$, a parameterized compositional mixture of feature adapters can be learned to identify the applied linguistic features for a given input and activate the corresponding feature adapters, thereby facilitating the effective addressing of linguistic discrepancies between SAE and its dialect variants.

To sum up, the **compositionality** of DADA enables targeted adaptation to specific dialect variants by selecting appropriate feature adapters. DADA uses modularity and compositionality to adapt a model to linguistic features present at test time since the pervasiveness of a feature can vary greatly based on its applicability and density (Demszky et al., 2021). This allows DADA to simultaneously adapt to various dialects by using a comprehensive set of feature adapters. We explore this property further in Sec. 5, using its interpretability to study individual feature adaptations utilized (see Sec. 5).

## 4 Multi-Dialect Adaptation

In this section, we demonstrate how DADA can enable the adaptation of an existing SAE model to multiple dialect variants, taking Multi-Genre Natural Language Inference (MNLI; Williams et al. (2018)) task as an example.

### 4.1 Experimental Setup and Evaluation

As described in Sec. 3.2, we train a feature adapter for each transformation rule from Ziems et al. (2023), the collection of lexical and morphosyntactic transformation rules between SAE and its dialect variants. In total, we train nearly 200 feature adapters for downstream use. Here, we demonstrate that these features can be flexibly composed in DADA to improve model performance across multiple dialects simultaneously. We evaluate on five representative dialects: AppE, ChcE, CollSgE, IndE, AAVE. We employ RoBERTa Base (Liu et al., 2019b) that has been finetuned on the original SAE MNLI training dataset as the backbone model.

For each transformation rule, we generate a synthetic dataset by applying only that specific transformation rule to each example in the original MNLI training dataset. We only retain examples that differ from the original example, i.e., examples that have been transformed. Afterward, we train feature adapters using these synthetic datasets, as described in Sec. 3.2. To aggregate trained feature adapters into the backbone model, we train a large fusion layer for 5 epochs on a synthetic dataset that applies all dialectal variations simultaneously, termed Multi. Additionally, we include a *null* adapter that remains as the identity function. This is kept for purely SAE inputs. In Appendix B, we report full hyperparameters along with the training details. We evaluate DADA on five English dialects: AppE, ChcE, CollSgE, IndE, AAVE and report the results in Table 1. Followed by Ziems et al. (2022, 2023), we construct each dialect-specific MNLI dataset by utilizing a subset of transformation rules that correspond to the respective dialect.

### 4.2 Results

Compared to the standard SAE model trained on the original MNLI dataset (SAE baseline), DADA demonstrates significant performance improvements across all evaluated dialects and even on SAE, with an average improvement of 2.16%. Moreover, DADA delivers comparable performance to the strong baseline provided by individual further fine-tuning or adapter tuning on the SAE trained model with dialect-specific training data (Single Finetuning and Single Adapter). However, while these two approaches require a perfect dialect identification system and $D$ models, our approach uses a single model and therefore does not rely on dialect identification. This makes DADA a simpler and more realistic method for use when the target dialect distribution is unknown.

Compared to additional finetuning or adapter tuning Multi on standard SAE model (Multi Finetuning and Multi Adapter), DADA brings an average improvement of 0.32% and 0.47%, respectively. Moreover, it tunes fewer parameters during a single training run compared to Multi Finetuning. We confirm that the empirically strong performance of DADA stems from the effective use of the correct individual feature adapters in Sec. 5.

Note that with DADA, in instances where a new dialect arises, the integration of this new dialect can be achieved through the identification of the

| Dialect Adaptation Details | | | | Evaluation Performance | | | | | | |
|---|---|---|---|---|---|---|---|---|---|---|
| Method | Dialect Data | Total Params. | Dialect Params. | AppE | ChcE | CollSgE | IndE | AAVE | Mean | SAE |
| SAE Baseline | - | 125$M$ | 0 | 83.70 | 84.91 | 80.62 | 82.00 | 83.95 | *82.71* | 86.57 |
| Finetuning | Multi | 125$M$ | 125$M$ | 85.72 | 86.33 | 85.00 | 85.09 | 84.44 | $85.30_{\pm0.31}$ | 86.72 |
| Adapter | Multi | 126$M$ | 1.5$M$ | 85.68 | 86.38 | 84.26 | 84.76 | 84.66 | $85.15_{\pm0.32}$ | 86.73 |
| Dada | Multi | 316$M$ | 192$M$ | 86.00 (+2.30) | 86.70 (+1.79) | 84.59 (+3.97) | 85.37 (+3.37) | 85.50 (+1.55) | $85.62_{\pm0.31}$ | 87.16 (+0.59) |
| Dada w/o null | Multi | 315$M$ | 190$M$ | 86.14 (+2.44) | 86.61 (+1.70) | 84.25 (+3.63) | 84.80 (+2.80) | 85.74 (+1.79) | $85.49_{\pm0.31}$ | 87.08 (+0.51) |
| Finetuning | single | $D \cdot 125M$ | $D \cdot 125M$ | 85.74 | 86.45 | 84.84 | 85.11 | 86.15 | *85.56* | 86.57 |
| Adapter | single | $D \cdot 126M$ | $D \cdot 1.5M$ | 86.23 | 86.53 | 84.85 | 85.40 | 86.26 | *85.63* | 86.57 |

Table 1: **Multi-Dialect Adaptation** results of SAE RoBERTa Base (Liu et al., 2019b) model for five English dialects: AppE, ChcE, CollSgE, IndE and AAVE. Due to the submission limitations of the GLUE benchmark, the results are reported on the validation mismatched split. The significance bars of the mean accuracies are determined through a paired bootstrap test conducted on the concatenation of each individual dialect dataset. $D$ is the number of target dialects for dialect adaptation. Dada outperforms the standard SAE baseline on all five dialects and SAE (marked as (+)), with an averge of 2.16% improvement. Most importantly, Dada achieves comparable performance and even surpasses (underlined) that of individual models.

linguistic transformation rules that govern the shift from SAE to the new dialect, followed by the training of a feature adapter for each new transformation rule, and finally the retraining of the fusion layer. Furthermore, the potential for reusability of trained feature adapters is significant as many dialects often share common linguistic features.

***null* adapter** For SAE inputs, every adapter has the potential to incorrectly change the model's original predictions. Therefore, we introduce a *null* adapter that which preserves the output of the original SAE model at each layer. We conduct an ablation study to evaluate the necessity of the *null* adapter by comparing with models where it is excluded. We denote this variant as Dada w/o *null*. As shown in Table 1, excluding the *null* adapter results in a slight drop in performance for SAE.

**Number of feature adapters** We analyze the average performance of Dada on 5 evaluated English dialects, considering different numbers of feature adapters ($k$) ranging from 1 to all. For each $k$, we select the top $k$ feature adapters with the best performance on the evaluation set. The results in Figure 3 demonstrate an overall increasing trend, indicating that each feature adapter incorporated in Dada can contribute to performance improvement, rather than relying solely on a select few.

## 5 Interpretability

As discussed in Sec. 3, Dada can implicitly identify the relevant linguistic features for a given input and activate the corresponding feature adapters. We validate this by investigating the correlation between attention scores within each layer of Dada and the presence of linguistic features, to determine whether the contributing feature adapters are relevant to the features present.

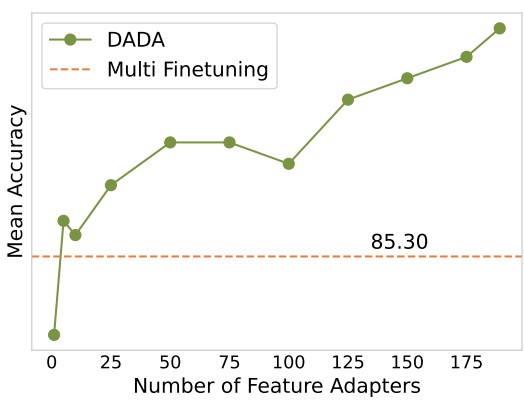

Figure 3: The mean accuracy of Dada shows an overall upward trend with the number of feature adapters.

### 5.1 Analyses Setup and Results

Here, we use the AAVE dialect and MNLI task as an example. To adapt a standard MNLI-finetuned RoBERTa Base model to target the AAVE dialect, we only need to take into account the 10 transformation rules between SAE and AAVE proposed by Ziems et al. (2022). We select the corresponding feature adapters from our collection and dynamically aggregate them by training a fusion layer on AAVE training set for 5 epochs with a learning rate 5e-5 and batch size 64. We evaluate the resulting model on the test split of the AAVE matched MNLI dataset as shown in Table 2. In comparison to the standard SAE model, Dada demonstrates a 3.2% and 1.4% improvement on AAVE and SAE, respectively. Moreover, Dada outperforms simple additional finetuning and adapter tuning of AAVE on SAE model by 0.4% and 0.5%, respectively, achieving the best performance of 86.6% on AAVE. These results demonstrate the superior performance of Dada over all other methods evaluated.

| Dialect Adaptation Details | | | Test Acc. | |
|---|---|---|---|---|
| Backbone | Method | Data | **AAVE** | **SAE** |
| *pretrained* | FT | **SAE** | 83.4 | 86.2 |
| | FT | **SAE** + **AAVE** | 84.8 | 85.6 |
| *SAE finetuned* | FT | **AAVE** | 86.2 | 87.4 |
| | Adapter | **AAVE** | 86.1 | 87.4 |
| | DADA | **AAVE** | 86.6✓ | 87.6 ✓ |

Table 2: **AAVE** **Adaptation** results of RoBERTa Base (Liu et al., 2019b). *pretrained* denotes the pretrained RoBERTa Base model, while *SAE finetuned* denotes the RoBERTa Base model that has been finetuned on the original **SAE** MNLI dataset. FT refers to "fine-tuning". DADA demonstrates superior performance on **AAVE** and **SAE** compared to baselines (marked as ✓).

## 5.2 Correlation Analysis of Fusion Activation

We perform a correlation analysis of these 10 feature adapters for the linguistic features applied to the input data. For each transformation rule, we calculate the softmax activation for each adapter, for each input to which the specific linguistic feature applies, and average over all activations within the same layer calculated over all instances in the **AAVE** MNLI test set. For better clarity, our final metrics takes the average utilization score of each feature adapter for the entire dataset and then subtracts the average utilization score associated with each transformation rule.

We plot the results for layers 1, 3, 7, 11 in Figure 4. We found that significant correlations in utilization on the lower layers (0-3) are observed, while those on the middle and higher layers are found to be negligible. This is consistent with our intuition, as the primary distinction between **SAE** and its dialect variants lies in their linguistic features (lexical and morphosyntactic), which are mainly captured by the lower layers of the model[5]. This analysis demonstrates that DADA has the capability to detect which linguistic features are relevant to the given input, and subsequently trigger the corresponding feature adapters. This highlights the interpretability of DADA with regard to the underlying factors that contribute to performance improvement.

## 6 Multi-Task Dialect Adaptation

Recent LLMs such as FLAN-T5 (Chung et al., 2022) and InstructGPT (Ouyang et al., 2022) are instruction-tuned (Wei et al., 2022) for various tasks, which is orthogonal to our method, making

it possible to combine the two approaches easily. In this section, we demonstrate how DADA can be employed to instruction-tuned LLMs to improve their task-agnostic performance on dialects.

## 6.1 Experimental Setup

Using **AAVE** dialect as a case study, to demonstrate the effectiveness of our method in adapting the **SAE** model across multiple tasks, we include the tasks from the **AAVE** transformed version (Ziems et al., 2022) of the GLUE Benchmark (Wang et al., 2018), including CoLA, MNLI, QNLI, QQP, SST-2, and STS-B. For our backbone model, we employ a FLAN-T5 Base (Chung et al., 2022). Despite the original paper incorporates GLUE within the FLAN-T5's training data, we retrain the model on these specific tasks to enhance its suitability.

## 6.2 Multi-task training

For each transformation rule of **AAVE** dialect, we construct synthetic training data following the procedure described in Sec. 3.1. However, in the case of a multi-task model, we construct a synthetic dataset for each task considered and utilize the mixture to train the corresponding feature adapter. Subsequently, we proceed to fuse these feature adapters by training a fusion layer on the super-synthetic dataset Multi-Task AAVE, which is constructed by applying all the **AAVE** transformation rules. In Appendix D, we provide the templates used to train the FLAN-T5 model. In Appendix B, we report full hyperparameters along with the training details. We assess the performance of DADA on **AAVE** transformed version of the GLUE Benchmark, and compare its results with the SAE baseline and Adapter Tuning with Multi-Task AAVE.

## 6.3 Results

It is surprising to note that although single Adapter Tuning with Multi-Task AAVE demonstrates improvements in 4 out of 7 tasks, the overall average performance is even inferior to that of the SAE baseline. In contrast, DADA consistently outperforms both the SAE baseline and Adapter Tuning across all evaluated tasks, resulting in an overall improvement of 1.80/1.92 points on the **AAVE** GLUE benchmark, respectively. Specifically, on the relatively large datasets, DADA achieves a notable accuracy improvement of 2.0%/1.0% on MNLI-mm, 0.9%/1.2% on QNLI, and 1.5%/0.9% on QQP when compared to the SAE Baseline and Adapter Tuning, respectively. These results demonstrate

---

[5]The linguistic feature differences are subtle. Although the absolute values of the correlation coefficients are not large, they are sufficient to indicate the existence of correlations.

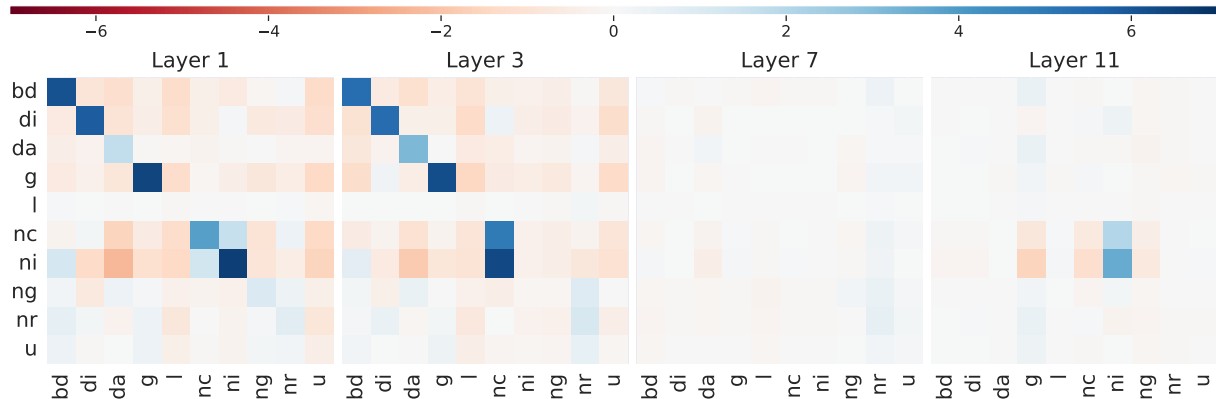

Figure 4: **Correlation Coefficients** for AAVE adaptation between the feature adapters (column) and the inputs to which specific linguistic features (row) apply in layers 1, 3, 7, 11. (See Appendix C for other layers.) Significant correlations can be observed within the lower layers (0-3), whereas there appear to be little to no correlations in the middle and higher layers. We use abbreviations for certain terms, such as "nc" for "negative_concord."

| | AAVE GLUE Performance | | | | | | | |
|---|---|---|---|---|---|---|---|---|
| Method | CoLA | MNLI-m | MNLI-mm | QNLI | QQP | SST2 | STS-B | Mean |
| SAE Baseline | 21.1 | 83.2 | 82.6 | 90.6 | 87.1 | 92.1 | 86.4 | 77.59 |
| Adapter Tuning | 18.2 | 84.1 | 83.6 | 90.3 | 87.7 | 92.9 | 85.5 | 77.47 |
| DADA | 26.3 ✓ | 84.4 ✓ | 84.6 ✓ | 91.5 ✓ | 88.6 ✓ | 93.7 ✓ | 86.6 ✓ | 79.39 ✓ |
| ChatGPT | 26.33 | 59.60 | 63.00 | 82.00 | 72.40 | 95.00 | 80.15 | 68.35 |
| ChatGPT + "Native Speaker" | 18.24 ↓ | 56.00 ↓ | 57.20 ↓ | 73.60 ↓ | 67.60 ↓ | 91.60 ↓ | 48.91 ↓ | 59.02 ↓ |

Table 3: **Multi-Task AAVE Adaptation** results of SAE FLAN-T5 Base (Chung et al., 2022) (Matthew's Corr. for CoLA; Pearson-Spearman Corr. for STS-B; Accuracy for all others). SAE Baseline denotes the FLAN-T5 Base model that has been finetuned using the original SAE mixture of task datasets. In comparison to both the SAE Baseline and Adapter Tuning with Multi-Task AAVE, DADA consistently exhibits superior performance across all evaluated tasks (marked with ✓). Due to budget constraints, the results of ChatGPT are reported on a randomly sampled 500 example subset of the development sets. Prompt based interventions do not improve ChatGPT's performance on AAVE. On the contrary, it can even result in further degraded performance (marked with ↓).

that our proposed approach, DADA, is not limited to single-task applications but can be easily scaled up to accommodate various tasks for use with the increasingly common multi-task instruction-tuning setup using in popular large-scale industrial systems (Ouyang et al., 2022; OpenAI, 2023a; Anil et al., 2023; OpenAI, 2023b).

In Table 3, we also present the results obtained with ChatGPT[6] (OpenAI, 2023a). Due to budget constraints, we were only able to evaluate randomly sampled 500 examples from the development set of each task. However, even with this limited evaluation, we can still gain insights that ChatGPT performs significantly worse than the SAE FLAN-T5 Base model on 5 out of 7 tasks. This emphasizes that merely scaling up the model is inadequate for tackling the challenge of dialect disparities. These limitations persist even in the context of large language models. Inspired by "*expert*"

prompts (Odena et al., 2021; Shi et al., 2022), we incorporate a "Native Speaker" Prompt for Chat-GPT:

"You are a native [DIALECT_NAME] English speaker, and here is your task:"

However, ChatGPT + "Native Speaker" Prompt does not yield improved results and, in fact, performs even worse than the vanilla ChatGPT on all evaluated tasks. This highlights that dialect adaptation is not solved with trivial prompt-based interventions while being simultaneously less grounded in expert linguistic resources than DADA.

## 7 Conclusion

In this paper, we present Dialect Adaptation via Dynamic Aggregation (DADA), a fine-grained and modular approach designed to adapt an established model trained on Standard American English to its dialect variants through the compositional aggregation of linguistic features. Our experiments demonstrate that the compositionality of DADA en-

---

[6]Engine: gpt-3.5-turbo. We conducted our ChatGPT experiments on May 16, 2023.

ables targeted adaptation to specific dialects, and demonstrated improved robustness across multiple evaluated dialects, including AppE, ChcE, CollSgE, IndE, and AAVE. Our analysis also highlights the interpretability of DADA, as shown through its capability to identify relevant linguistic features for a given input and trigger the corresponding adapters. Furthermore, our experiments on FLAN-T5 illustrate the potential of applying DADA to task-agnostic instruction-tuned large language models, showcasing its generalizability.

## Limitations

DADA involves the training for feature adapters and the fusion layer, which can make it computationally expensive, especially when dealing with a substantial number of linguistic rules. However, each training run only requires a small number of parameters to be learned, and parallelization is feasible for feature adapter training. More importantly, these trained feature adapters exhibit significant reusability; the same set of feature adapters can be reused and employed for multiple dialects, though the fusion layer would need to be retrained for these dialects. However, if a use case does not involve significant reuses, this aspect may indeed remain a limitation. We will release our trained feature adapters so that future studies will not need to reincur the up-front training cost.

Furthermore, while DADA has the flexibility to utilize any linguistic rules, in our experiments, we specifically employed these linguistic transformation rules that are well-established in prior work for English (Ziems et al., 2022, 2023). These rules were chosen because they were curated by linguists, validated by dialect speakers, and because English has many globally relevant dialects (Bird, 2022). However, evaluating DADA for other language groups and broader sets of lexical variation is key area for future work.

While DADA mainly relies on Multi-VALUE (Ziems et al., 2022, 2023), they are orthogonal processes with different assumptions about dialect use. For each dialect, Multi-VALUE defines the density of a dialectal feature as the probability of the feature occurring when it is applicable, as well as the probability of the corresponding perturbation to be used in converting a sentence from SAE into that dialect. However, the actual prevalence of a feature heavily depends also on applicability.

DADA instead focuses on adapting to the lin-guistic features present in a given sentence. We learn a parameterized compositional mixture of the dialectal features automatically, rather than relying on static assumptions of density. This avoids what we view as a major issue: it is often difficult to determine the dialect of an input since dialects themselves vary depending on context and speaker. The density of a dialectal feature represents an approximate of density across the entire dialect, but may not be accurate to a specific speaker and context (Koenecke et al., 2020). On the other hand, DADA can dynamically recognize the applicable dialectal features for a given input and activate the corresponding feature adapters. It remains to be explored in future work how the density of dialectal features, as captured in the linguistic literature, relates to the compositional mixture of these features as learned in the fusion layer of DADA.

## Ethics Statement

Previous linguistic works on dialectal features may not fully or accurately document the natural usage patterns of all existing dialects in terms of their linguistic rules. As a result, we acknowledge that our proposed method DADA, which relies on these dialectal features from prior literature, may not take some undocumented features associated with dialects into account. However, by curating more dialectal features, our method can be easily extended to a broader range of dialects. Additionally, as DADA is task-agnostic when applied to instruction-tuned models (Sec 6), malicious individuals might misuse it. To address this concern, we will release DADA with a license that explicitly prohibits its usage for purposes of deception, impersonation, mockery, discrimination, hate speech, targeted harassment, and cultural appropriation targeting dialect-speaking communities.

## Acknowledgement

We would like to thank the anonymous reviewers and SALT lab members for their valuable feedback. This work was partially sponsored by the Defense Advanced Research Project Agency (DARPA) grant HR00112290103/HR0011260656, and NSF grant IIS-2247357 and IIS-2308994.

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

## A    Tranformation Rules Details

Ziems et al. (2022, 2023) developed a collection of lexical and morphosyntactic transformation rules that account for the differences in linguistic features between SAE and its various dialect variants. In our study, we build upon this work by training transformation adapters for each rule in this collection. In their original paper, they present a comprehensive overview of each transformation rule in Appendix B. In Tables 9-21, they provide detailed Multi-VALUE implementations, including an enumeration of the implemented dialects and features, accompanied by illustrative examples for each.

Furthermore, we provide detailed statistics for the respective synthetic training datasets (for MNLI task) associated with each linguistic rule for the AAVE dialect in Table 4. While we do not present statistics for every linguistic feature for all dialects across all evaluated tasks, we release our code, all synthetic datasets and the trained adapters, to further improve the reproducibility.

| Linguistic Rule | Size | Eval Acc |
|---|---|---|
| been_done | 48,515 | 84.46 |
| dey_it | 33,927 | 84.41 |
| drop_aux | 78,157 | 84.06 |
| got | 25,203 | 83.41 |
| lexical | 331,784 | 86.11 |
| negative_concord | 49,529 | 84.41 |
| negative_inversion | 658 | 83.03 |
| null_genetive | 50,122 | 84.11 |
| null_relcl | 45,899 | 83.70 |
| uninflect | 124,447 | 84.64 |

Table 4: Linguistic Rules, Dataset Size, and Feature Adapter Accuracy for AAVE dialect (MNLI task).

## B    Training Details

**Multi-Dialect Adaptation** We train feature adapters for each transformation rule using synthetic datasets, as described in Sec. 3.2, with learning rate 3e-4 and batch size 64 followed by Houlsby et al. (2019). To prevent significant performance differences among the trained feature adapters due to varying sizes of synthetic datasets, we fix the number of training steps to 10,000. For each feature adapter, we choose the checkpoint with the highest accuracy on the validation matched split of a synthetic dataset that applies all dialectal variations simultaneously, termed `Multi`. For dynamic

aggregation, we train a large fusion layer for 5 epochs on `Multi`. We set the learning rate to 2.5e-5 and the batch size to 64.

**Multi-Task Dialect Adaptation** For feature adapter training, we set the learning rate to 1e-3 and fix the number of training steps as 50000. To fuse these feature adapters, we train a fusion layer for 5 epochs using a learning rate of 8e-5.

Throughout the process of model training (including finetuning, adapter tuning, DADA training etc.), we consistently employ the standard training objectives specific to the tasks, such as cross-entropy loss for classification tasks.

## C  Utilization Correlation Coefficients Plots

In Sec. 5, we showcase the effectiveness of DADA in adapting the *RoBERTa Base* (Liu et al., 2019b) model that has been finetuned on the original **SAE** MNLI training dataset to **AAVE**. To demonstrate the interpretability of DADA, we conduct an analysis of the utilization correlation among the aggregated 10 transformation adapters. We present utilization correlation coefficient plots for all layers in Figure 5 and 6.

## D  FLAN-T5 Templates

We provide here the templates used in Sec. 6 to train the FLAN-T5 model for each task. In the original paper by Chung et al. (2022), they defined 10 templates for each task and randomly applied them to each training example to enhance the model's robustness to varying instruction wordings. However, in our study, our goal is to demonstrate the generalizability of our proposed method DADA to instruction-tuned models, rather than focusing on improving the model's instruction-following capability. Therefore, for each task, we fix the usage of the first template from the set of 10 templates designed in the original paper.

**CoLA**  The Corpus of Linguistic Acceptability (CoLA; Warstadt et al. (2018)) task is a widely used benchmark that focuses on grammatical acceptability judgments. It aims to assess the ability of models to determine whether a given sentence is syntactically and semantically correct or not. For the CoLA task, we adopt the following template:

> Sentence: {sentence}
> Would a linguist rate this sentence to be acceptable linguistically?
>
> I think the answer is {answer}

**MNLI**  The Multi-Genre Natural Language Inference (MNLI; Williams et al. (2018)) is designed to assess the model's ability to comprehend and reason. MNLI involves determining the logical relationship - entailment, contradiction, or neutrality - between a given premise and a corresponding hypothesis. For the MNLI task, we adopt the following template:

> Premise: {premise}
>
> Hypothesis: {hypothesis}
>
> Does the premise entail the hypothesis?
>
> {answer}

**QNLI**  The Question-answering Natural Language Inference (QNLI; (Wang et al., 2018)) task is a prominent benchmark that focuses on assessing

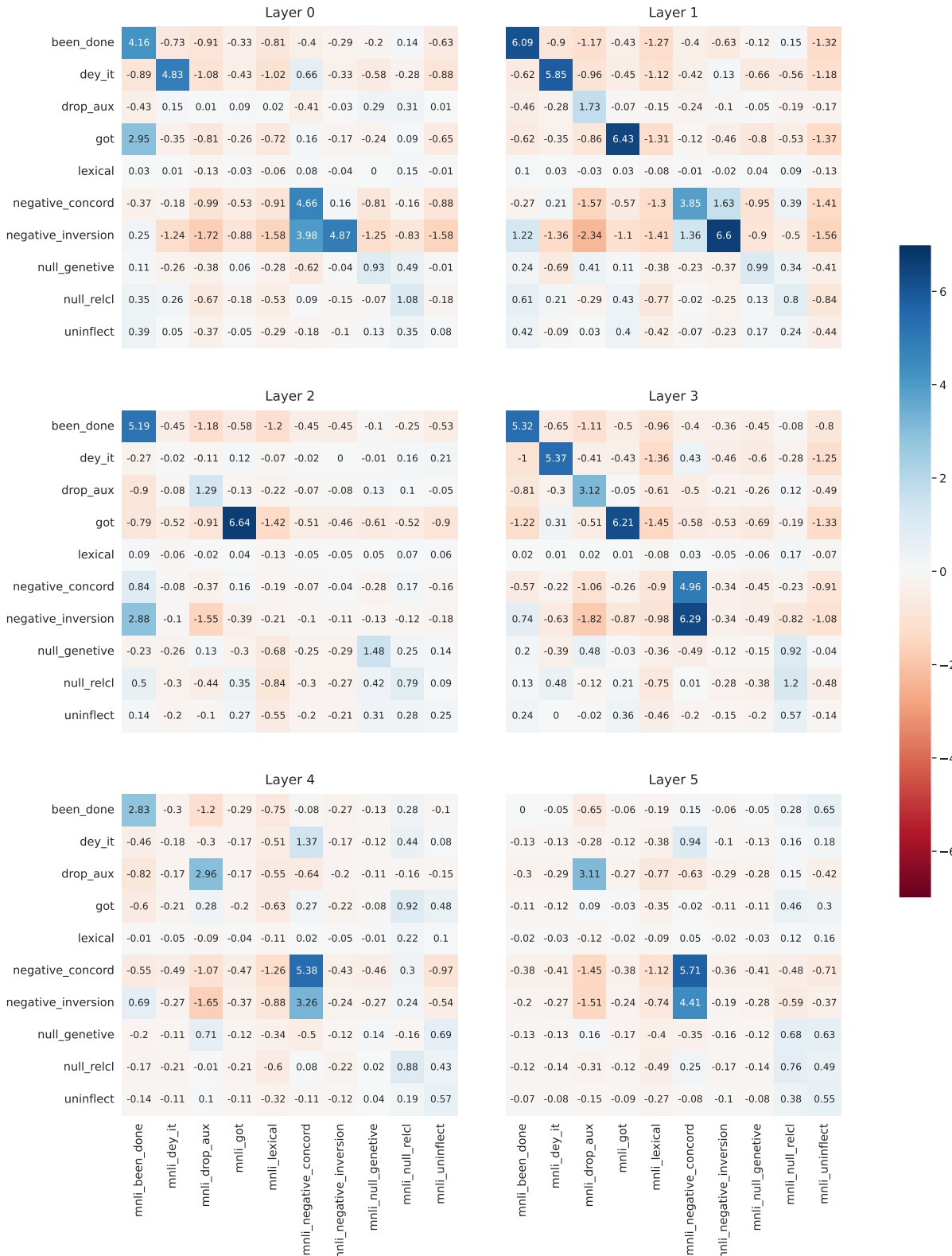

Figure 5: **Correlation Coefficients** between the transformation adapters (column) and the inputs to which specific transformation rules (row) apply in layers 0-5.

the ability of models to perform sentence-level semantic matching and reasoning. In this task, given a question and a corresponding sentence, the objec-tive is to determine whether the sentence contains the answer to the question, considering both lin-guistic and logical entailment. For the QNLI task,

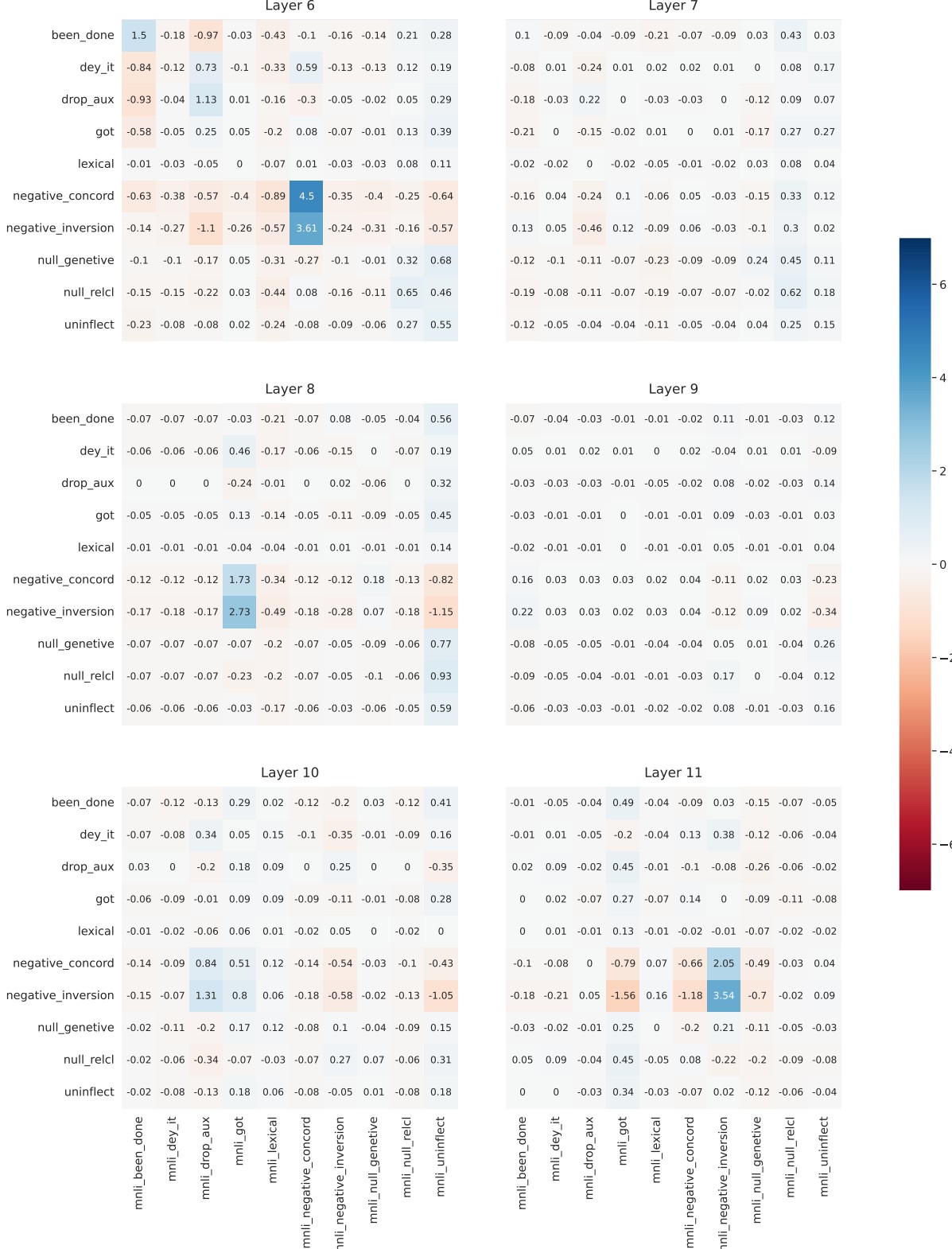

Figure 6: **Correlation Coefficients** between the transformation adapters (column) and the inputs to which specific transformation rules (row) apply in layers 6-11.

we adopt the following template:

> Does the sentence **{sentence}** answer the question **{question}**
>
> **{answer}**

**QQP** The Quora Question Pairs[7] (QQP) task is a widely recognized benchmark that focuses on question sentence similarity. The task involves determining whether a pair of questions asked on the Quora platform is semantically equivalent or not. For the QQP task, we adopt the following template:

> {question1}
> {question2}
> Would you say that these questions are the same?
> {answer}

**SST-2** The SST-2 (Stanford Sentiment Treebank; Socher et al. (2013)) task is a widely used benchmark for sentiment analysis. It involves classifying the sentiment of a given sentence as either positive or negative. For the SST-2 task, we adopt the following template:

> Review:
> {sentence}
> Is this movie review sentence negative or positive?
> The answer is: {answer}

**STS-B** The Semantic Textual Similarity Benchmark (STS-B; Cer et al. (2017)) task is a widely recognized benchmark that evaluates the ability of models to assess the semantic similarity between pairs of sentences. The task involves assigning a similarity score to pairs of sentences based on their semantic equivalence. For the STS-B task, we adopt the following template:

> {sentence1}
> {sentence2}
>
> Rate the textual similarity of these two sentences on a scale from 0 to 5, where 0 is "no meaning overlap" and 5 is "means the same thing".
>
> {answer}

---

[7] https://www.kaggle.com/c/quora-question-pairs