# OpenReview forum: "DADA: Dialect Adaptation via Dynamic Aggregation of Linguistic Rules"
_EMNLP/2023/Conference — EMNLP 2023 Main_

### Official Review · Reviewer_XXyA · 2023-08-04

**Soundness:** 4

**Excitement:**

4: Strong: This paper deepens the understanding of some phenomenon or lowers the barriers to an existing research direction.

**Paper Topic And Main Contributions:**

In this paper, the authors propose **DADA**, a flexible and efficient framework of language model adaptation to non-standard (non-SAE) dialects of English. They train individual adapters corresponding to linguistic features that are characteristic of various dialects, and ensemble them using AdapterFusion in order to improve language model performance on the target varieties. Using various tasks from the GLUE benchmark, including MNLI and SST2, the paper shows that DADA can outperform the baseline of fine-tuning or adapter-tuning a language model such as RoBERTa and FLAN-T5 on SAE data on a range of dialects, for instance, African-American Vernacular English, Appalachian English, and Indian English.

It is true that the performance improvements are relatively minor, and that DADA generally underperforms the baseline of training adapters on dialectal datasets. However, the authors' method circumvents the need for reliable dialect-identification which is far from a solved issue. Additionally, DADA is a highly interpretable method where the activation of adapters corresponding to individual linguistic features can be indicative of the features present in a target text or variety. Furthermore, DADA has the potential to inspire a wide range of follow-up papers experimenting with feature adapters for language model adaptation for multilingual NLP and varieties of other languages.

Overall, it is a strong paper that I would recommend for acceptance.

**Questions For The Authors:**

A: Can we be certain that the feature adapters adequately learn to adapt to the linguistic features they should? What about cases when there are very few training instances due to the rarity of a certain features? What about when the difference is very subtle between "dialectal" and standard sentences?

B: The authors analyse the impact feature adapters have on the mean accuracy on the MNLI task. However, despite their claim that there is an overall increasing correlation with the number of adapters added, performance markedly decreases between 75 and 100 adapters. Is there something specific about the adapters added in that stage? Additionally, the figure also does not have numbers on the y axis, making it difficult to read the exact impact of the adapters.

C: When testing the various setups on MNLI, the authors use a so-called "validation mismatched split" to evaluate on. Can they clear up what they mean by that?

D: What is the training objective the various feature adapters are trained on? What data sizes are used for training the individual adapters?

**Reasons To Accept:**

* The paper proposes a promising methodology which is novel and uses linguistic information for dialectal adaptation.
* It reasons well for the need for such adaptation on ethical, moral and societal grounds.
* It reasons well for the advantages of its methodology in terms of difficulties of dialectal identification which makes targetted adaptation to dialects challenging.
* It provides a extensive literature review on dialect adaptation and parameter-efficient learning.
* Its framework is flexible enough to allow adaptation to a range of potential target dialects across a number of various tasks.
* The aspects of interpretability are promising.

**Reasons To Reject:**

* The authors compare their DADA with other setups across dialects only on a single task, MNLI, and reserve the other tasks for a single dialect. This might limit how generalisable the results are.

* It would be an asset if the authors could release the feature adapters to the public, currently this is missing.

* Relative efficiency of the method compared to maintaining individual dialectal adapters is questionable. There are presumably much fewer English dialects than linguistic features that set them apart and it is not true that for fine-tuning adapters we would need as many separate models as many dialects.

* There are concerns with reproducibility of the paper. The authors do not report what training objective they use to create their feature adapters and the size of the training datasets for each. They also do not state whether they carry out multiple experimental runs to make their results more robust. Finally, they compare DADA on various GLUE tasks against ChatGPT. Due to limitations, they limit the development set to 500 examples, but do not state either the version number of ChatGPT, nor the basis on which they make this selection.

**Reproducibility:**

4: Could mostly reproduce the results, but there may be some variation because of sample variance or minor variations in their interpretation of the protocol or method.

**Reviewer Confidence:**

4: Quite sure. I tried to check the important points carefully. It's unlikely, though conceivable, that I missed something that should affect my ratings.

---

> ### Author Rebuttal · Authors · 2023-08-27
>
> Thank you for your thorough review of our paper. We extend our gratitude for your thorough and insightful evaluation of our paper. And we greatly appreciate your recognition of the most significant advantage of DADA: enhancing multi-dialect robustness without relying on a high-accuracy dialect-identification system, a challenge that is still far from being resolved. We also greatly appreciate your interest in applying our DADA to languages other than English and multilingual NLP.
>
> We agree that evaluating **robustness** across dialects and across tasks simultaneously would solidify our confidence in the generalizability of our method. However, we believe that our current two experiments are complementary, and when combined, there is sufficient evidence to demonstrate that DADA can enhance multi-dialect robustness across multiple tasks. And to further strengthen this, we will release our code and all our trained feature adapters to the public later and this will enable people to verify by themselves.
>
> Regarding the **efficiency** of DADA compared to individual dialect adapters, it's crucial to acknowledge that the latter approach assumes the presence of a highly accurate dialect identification system and a prior knowledge of the specific dialect associated with the input. However, this assumption is not feasible in real-world scenarios. Furthermore, flexible boundaries and the evolution of dialects means that while feature adapters could be reusable, fixed dialect adapters might need more frequent retraining.
>
> Regarding the **reproducibility**, we provide additional experimental details as follows: Throughout the process of model training (including finetuning, adapter tuning, DADA training etc.), we consistently employed the standard training objectives specific to the tasks, for example cross-entropy loss for classification: $- \sum_{i=1}^{N} y_i \cdot \log(p_i)$. In order to ensure robustness, we conducted a total of 5 runs for each experiment (except for time or resources intensive ones). The reported final results were based on the average values derived from these runs. Regarding the experiments conducted with ChatGPT, all were executed with model "gpt-3.5-turbo" on May 16 2023, and for each task, we randomly sampled 500 examples for the evaluation.
>
> Regarding the **statistics** of the training datasets, we had a table depicting the size of each training dataset, which we later opted to remove due to space constraints. For your reference, here are some statistics regarding the training datasets for AAVE adaptation.
>
> [Table: Linguistic Rules, Dataset Size, and Feature Adapter Accuracy for AAVE Dialect]
> | Linguistic Rule      | Size    | Eval Acc |
> |----------------------|---------|----------|
> | been_done            | 48,515  | 84.46    |
> | dey_it               | 33,927  | 84.41    |
> | drop_aux             | 78,157  | 84.06    |
> | got                  | 25,203  | 83.41    |
> | lexical              | 331,784 | 86.11    |
> | negative_concord     | 49,529  | 84.41    |
> | negative_inversion   | 658     | 83.03    |
> | null_genetive        | 50,122  | 84.11    |
> | null_relcl           | 45,899  | 83.70    |
> | uninflect            | 124,447 | 84.64    |
>
> Unfortunately, due to limitations in space, we are unable to present the statistics for all the linguistic features. However, we plan to re-include this information in the appendix of the revised version of our paper. To further improve the reproducibility, we are committed to release our code and all the trained adapters later.
>
>
>
> Below are the responses to your supplementary questions:
>
> -**QA:** For DADA, the availability of training instances is not an issue, even though the linguistic features are quite infrequent. This is because the training samples we utilize are synthetic data - dialect text transformed from standard text. And it has been shown that DADA can deal well with very subtle linguistic features. Among the linguistic features we are currently considering, there are already some very subtle linguistic features, and DADA has adapted to them quite effectively. For example, within the linguistic feature 'regularized_reflexives,' the difference is often only a letter (e.g., 'myself' and 'meself').
>
> -**QB:** We don’t think anything particularly noteworthy happened here; it is possible that this is merely some noise. Also the decrease is only minor and not significantly marked. And because our experimentation involved only every 25 adapters, experiments with finer granularity may help diminish such noise.
>
> -**QC:** There are two validation sets of MNLI - “mismatched” and “matched”. The "mismatched" split means that the text comes from a different distribution than the training set. Following previous works, we evaluate the model on the validation matched set during training and report the final performance on the mismatched set.
>
> -**QD:** See above.
>
> Furthermore, we are committed to releasing all the code of our paper, as well as making all of our trained adapters available as assets for the entire community to use, which also stands as a main contribution of our work.

---

### Official Review · Reviewer_KiYC · 2023-08-04

**Soundness:** 3

**Excitement:**

4: Strong: This paper deepens the understanding of some phenomenon or lowers the barriers to an existing research direction.

**Paper Topic And Main Contributions:**

The paper presents an approach to dialect adaptation of a number of English variants. It demonstrates that through training features adapters they can improve the performance of language models. The paper also claims that introducing new dialects can be achieved by identifying the linguistic transformation rules from SAE to the new dialect.

**Reasons To Accept:**

The paper presents an interesting approach to dialect adaptation, and the results of the mentioned experiments are promising. I would be curious to see this approach applied to other languages than English.

**Reasons To Reject:**

The dynamic aggregation process relies on having a set of lexical and morphosyntactic rules, but there was no mention of how these rules are created. Are they manually curated or automatically generated from existing data, since if this is a manual process it will present an expensive entry threshold.

**Reproducibility:**

3: Could reproduce the results with some difficulty. The settings of parameters are underspecified or subjectively determined; the training/evaluation data are not widely available.

**Reviewer Confidence:**

3: Pretty sure, but there's a chance I missed something. Although I have a good feel for this area in general, I did not carefully check the paper's details, e.g., the math, experimental design, or novelty.

---

> ### Author Rebuttal · Authors · 2023-08-27
>
> Thank you for your thorough review of our paper. We greatly appreciate your interest in applying our DADA to languages other than English. DADA's potential extends far beyond English and there is great potential in utilizing DADA for other languages. We think broad support of linguistic variation like dialects is key to supporting global and diverse speaker populations, making methods like DADA extremely valuable.
>
> Additionally, it's important to highlight that we have indicated in our paper the sources of all the linguistic features we are currently considering - these features are curated by linguists (eWAVE: https://ewave-atlas.org/) and are implemented as transformation rules by Multi-Value [1].
>
> It is notable that the evolution of languages/dialects is a slow process, hence the infrequent emergence of new linguistic rules. And the curation of these rules is a one-time effort; once curated, they can be consistently employed. Furthermore, we should emphasize that linguistic rule curation is much more efficient than constructing distinct datasets for every dialect variant.
>
> However, we recognize that this is still a complex and costly process. Therefore, we are currently engaged in a new project where we are leveraging LLMs to assist in this process. We're exploring ways in which LLMs can aid linguists or autonomously identify new linguistic rules from text corpora, aiming to streamline and enhance this process.
>
> Furthermore, we are committed to releasing all the code of our paper, as well as making all of our trained adapters available as assets for the entire community to use, which also stands as a main contribution of our work.
>
> [1] Caleb Ziems, William Held, Jingfeng Yang, Jwala Dhamala, Rahul Gupta, and Diyi Yang. 2023. Multi-VALUE: A Framework for Cross-Dialectal English NLP. ACL 2023.

---

### Official Review · Reviewer_ybzc · 2023-08-05

**Soundness:** 3

**Excitement:**

4: Strong: This paper deepens the understanding of some phenomenon or lowers the barriers to an existing research direction.

**Missing References:**

The bibliography is to be checked for uncapitalized proper names (like english) and incomplete references (like like 923).

**Paper Topic And Main Contributions:**

This paper focuses on dialectal robustness and puts forward an approach to handle specific linguistic features of Standard American English dialects by composing adapters. The methodology is compelling and the findings are very interesting.

**Reasons To Accept:**

I'd like this paper to be accepted because it goes beyond the current "macro-language" perspective in NLP and delves into dialects and non-standard varieties. Relevant studies (also cited in the paper) focus on the transformation of linguistic rules and various approaches using adaptors. Although the current study is very much in line with the previous work, it also generates synthetic data and dynamically aggregates the feature adapters which enhances the model’s multitasking and multi-dialect abilities.

**Reasons To Reject:**

Although the study is thorough and supports all its claims, it would have been interesting to have some qualitative analysis on the dialects and the linguistic features that make them different in the fine-tuned models.

**Reproducibility:**

3: Could reproduce the results with some difficulty. The settings of parameters are underspecified or subjectively determined; the training/evaluation data are not widely available.

**Reviewer Confidence:**

4: Quite sure. I tried to check the important points carefully. It's unlikely, though conceivable, that I missed something that should affect my ratings.

---

> ### Author Rebuttal · Authors · 2023-08-27
>
> Thank you for your thorough review of our paper. We extend our gratitude for your insightful evaluation of our paper and your recognition of **its distinctions from the prevailing "macro-language" perspective in NLP**, underscoring the uniqueness of our approach DADA. Thank you for identifying the unique perspective and contributions this work takes!
>
> In addition, we are grateful for your suggestion to include some more **qualitative analysis**. In Fig. 4, we already plotted the correlation coefficients between the feature adapter activations and the linguistic features for AAVE dialect. But as you suggested, it would be interesting to extend the analysis to the multi-dialect setting. We have supplemented this analysis, where we show the activations for each feature adapter across different dialects, and analyze their correlation with the pervasiveness of each linguistic feature within different dialects. Here are some results:
> | Dialect   | Pearson Coefficient |
> |-----------|----------------------|
> | AppE      | 0.1968               |
> | ChcE      | 0.2016               |
> | CollSgE   | 0.2724               |
> | IndE      | 0.2610               |
> | AAVE      | 0.2148               |
> As we mentioned in the paper, because linguistic feature differences are very subtle, and, in this analysis, we incorporated all of the nearly 200 linguistic features that we discuss in the paper. Although the correlation coefficient values are not large, they are sufficient to provide evidence to demonstrate that DADA can indeed identify linguistic features within the input text, activate the corresponding feature adapters, and thereby allow it to accommodate different dialects. Additionally, we will incorporate supplementary results and plots into the revised version of the paper.
>
> And finally, we will meticulously review and correct any existing typos to ensure our bibliography's correctness. And we are committed to releasing all the code of our paper, as well as making all of our trained adapters available as assets for the entire community to use, which also stands as a main contribution of our work.

---

### Meta-Review · Area_Chair_xuoN · 2023-09-17

**Recommendation:** 4

**Metareview:**

This paper presents a dialect adaptation to a number of English dialects without the need of a high-accuracy dialect identification systems. They propose DADA, a modular approach to adapt Standard American English (SAE) trained models to different dialect variants by composing adapters which handle specific linguistic features obtained from several linguistic rules. They trained on nearly 200 feature adapters that distinguish SAE and other English dialects.

All the reviewers appreciated the incorporation of the linguistic rules to the model, and the promising experiments, and the modularity of adapters incorporated in the model. There are a few concerns about how they obtained the 200 linguistic features which the authors adequately responded to. Another issue was the small improvement in performance. Overall, this is a great contribution for incorporating other english dialects.

---

### Decision · Program_Chairs · 2023-10-07

**Decision:**

Accept-Main

**Comment:**

This paper presents a dialect adaptation to a number of English dialects without the need of a high-accuracy dialect identification systems. They propose DADA, a modular approach to adapt Standard American English (SAE) trained models to different dialect variants by composing adapters which handle specific linguistic features obtained from several linguistic rules. They trained on nearly 200 feature adapters that distinguish SAE and other English dialects.

All the reviewers appreciated the incorporation of the linguistic rules to the model, and the promising experiments, and the modularity of adapters incorporated in the model. There are a few concerns about how they obtained the 200 linguistic features which the authors adequately responded to. Another issue was the small improvement in performance. Overall, this is a great contribution for incorporating other english dialects.